# Molecular Mechanisms of Diabetic Kidney Disease

**DOI:** 10.3390/ijms23158668

**Published:** 2022-08-04

**Authors:** Jorge Rico-Fontalvo, Gustavo Aroca, Jose Cabrales, Rodrigo Daza-Arnedo, Tomas Yánez-Rodríguez, María Cristina Martínez-Ávila, Isabella Uparella-Gulfo, María Raad-Sarabia

**Affiliations:** 1Colombian Nephrology Association, Bogotá 110221, Colombia; 2Management of Technologies and Innovation, Department of Engineering, Universidad Simón Bolivar, Cl. 58 #55-132, Barranquilla 080002, Colombia; 3Faculty of Medicine, Universidad Simón Bolívar, Barranquilla 080002, Colombia; 4Nephrology Fellow, Stanford University School of Medicine, Palo Alto, CA 94305, USA; 5Internal Medicine, Universidad de Cartagena, Cartagena 130001, Colombia; 6BIOTOXAM Group, Universidad de Cartagena, Cartagena 130001, Colombia; 7General Medicine, Universidad del Sinú, Cartagena 130015, Colombia; 8Internal Medicine, Universidad del Sinú, Cartagena 130015, Colombia

**Keywords:** genetics, epigenetic, inflammatory, innate, adaptive, cytokines, innovation

## Abstract

The inflammatory component of diabetic kidney disease has become of great interest in recent years, with genetic and epigenetic variants playing a fundamental role in the initiation and progression of the disease. Cells of the innate immune system play a major role in the pathogenesis of diabetic kidney disease, with a lesser contribution from the adaptive immune cells. Other components such as the complement system also play a role, as well as specific cytokines and chemokines. The inflammatory component of diabetic kidney disease is of great interest and is an active research field, with the hope to find potential innovative therapeutic targets.

## 1. Introduction

Diabetic kidney disease (DKD) is seen in approximately 30 to 40% of patients with diabetes mellitus (DM) type 1 and 2 worldwide [1]. Its pathogenesis and progression are distinguished by three main components, which include the hemodynamic, metabolic, and inflammatory components; the latter has become of interest given its prognostic and therapeutic implications. The early diagnosis of diabetic kidney disease is fundamental to slowing down the progression of worsening kidney function and to decreasing its poor prognosis. The genomic and molecular mechanisms could help in its early detection.

In this setting, the inflammatory and immune response play a central role in the pathogenesis of diabetic kidney disease [2]. The disease has been traditionally seen as a non-inflammatory glomerular condition, induced primarily by metabolic and hemodynamic changes derived from the chronic exposure to hyperglycemia. This has changed considering the new scientific evidence available, with inflammation being considered a main phenomenon for the initiation and progression of the disease [3].

RNA sequencing studies of the nuclei of renal cell biopsies of patients with type 2 DM support the activation of signaling pathways involved in inflammation [4,5]. Additionally, the diabetic kidney shows an increase in inflammatory cells, with leukocytes being seven to eight times more frequent in number than in healthy kidneys; there is evidence of monocytes, B cells, and plasma cells [4].

Several studies have contributed to the development of molecular pathways in DKD, focusing on the role of innate immunity and on the different inflammatory mediators such as cytokines, especially interleukin 1 (IL-1), IL-6, IL-18, and different cell populations [3]. 

The role of other cells such as monocytes and macrophages are important as well, and its activation has been considered a risk factor for the development of DKD [6]. The different inflammatory mechanisms involved in the pathogenesis of DKD are reviewed here, as well as potential diagnostic and future therapeutic targets. 

## 2. Genetics and Epigenetics of DKD

Not all patients with DM develop DKD; even with poor diabetic control, some patients do not develop kidney issues, indicating that additional factors are involved in the start and progression of the disease [7,8]. Several publications of genome-wide association studies (GWAS) demonstrated the role of genetic variants in the development of DKD [7,9].

The rise and growth of genetic and epigenetic studies in the development of DKD have aimed to better understand the pathophysiology of the disease [10]. Clinical and epidemiological studies have described the familial aggregation of DKD in several ethnic groups, demonstrating the role of genetics in the development of the disease [10,11]. There is a clear relationship between genetic, epigenetic, and environmental factors in the pathogenesis of DKD [10].

The process of DNA methylation, where the methyl group of S-adenosylmethionine is transferred to the cytosine of DNA under catalysis of DNA methyltransferases (DNMT), promotes the activation of immune cells in diabetic kidney disease [12]. It has been shown that DNA methylation is associated with kidney injury and inflammation in patients with diabetic kidney disease [12]. Inappropriate DNA methylation of the upstream regulators of the mammalian target of rapamycin (mTOR) leads to inflammation by promoting the upregulation of DNMT1 in patients with DKD [12].

Similarly, histone post-translational modifications also play a role in the pathogenesis of DKD, such as histone methylation, acetylation, and deacetylation. Chromatin histone acetylation is promoted by hyperglycemia, which enhances inflammatory gene expression; the presence of Sirt6, a histone deacetylase, exacerbates inflammation in the kidney [12]. Several non-coding RNAs have also been studied, including microRNA (miRNA), lcnRNA, and circRNA, all of which promote inflammatory gene expression and inflammation in the diabetic kidney [12].

Genetic studies of DKD have focused on the evaluation of genomic variants of DNA (for example, single nucleotide polymorphisms, copy number variants, and microsatellites) and different phenotypes of the disease [10]. The latter, alongside environmental factors, would explain the existing relationship between external factor exposure and genomic variants in susceptible populations. Current genetic studies of DKD have focused on large-scale genomic studies (known as GWAS and EWAS) [10]. In a GWAS of patients with type 2 DM with European ancestry, there was a specific locus identified as PLCB4, which was associated with the development of CKD; GABRR1, a novel locus, was associated with microalbuminuria in patients with European ancestry (not found in patients of Asian ancestry), but there is need for replication studies [13]. The expression of GABRR1 is upregulated in biopsies of patients with DKD [13].

## 3. Fundamentals of Metabolomics and Proteomics in Kidney Disease

The central dogma of molecular biology establishes the transference of information from DNA (genome) to RNA (transcriptome) to protein (proteome), and from there to metabolites (metabolome) [9]. DNA mutations (genomic changes) are fundamental for the development of diseases such as cancer, with less impact in kidney disease [9]. On the contrary, the behavior of the proteome and the metabolome is very dynamic and is fundamental to establishing the functional consequences of the changes in the genome. This is also referred to as “functional genomics”.

The advantage of analyzing the proteomics and metabolomics is that they can provide more specific information, as proteins and metabolites vary among tissues and different cell types [9]. For kidney disease, these include metabolites such as urea, uric acid, glucose, and creatinine and proteins such as cystatin C, complement, and albumin, which are very common in the practice of nephrologists. These proteins and metabolites are modifiable, which can be amenable to therapeutic targeting [9].

Though the study of the proteome and the metabolome has different strengths, they have a clear disadvantage compared to the genome, which is the inability to infer causality in relation to specific metabolic products [9].

In general, changes in genomics and DNA sequencing precede the development of the disease. However, when studying the origin of diseases, genomics tends to be unidirectional; different from the natural dynamics of the proteome and metabolome, which can be bidirectional, from metabolites to disease and vice versa [9]. The bidirectional nature may be of use in the early detection of disease, although this may be challenging [9]. An example of the use of proteomics is the detection of the M type phospholipase A receptor (PLA2R), which was the unknown antigen of membranous nephropathy, which since then has changed our diagnostic and therapeutic approach [9].

Early diagnosis of DKD is fundamental to decreasing its negative impact in the loss of kidney function [11,14]. However, conventional diagnostic testing usually leads to late diagnosis; this is the reason why metabolomics may reveal complex metabolic networks that may give additional information regarding the pathophysiology of DKD, which eventually can lead to the identification of unique metabolic pathways with distinctive characteristics and potential therapies [11]. Table 1 reviews animal experiments in biomarkers used for the study of kidney diseases or DKD, whereas Table 2 lists human trials on several biomarkers in patients with DKD. 

## 4. Toll-like Receptors and Nod-like Receptors

Membrane toll-like receptors (TLR) and cytoplasmic nod-like receptors (NLR) are the main sensors in immune cells and play a fundamental role in the initiation of the innate immune response by detecting the pathogen-associated molecular patterns (PAMPs) and damage-associated molecular patterns (DAMPs) [3].

TLRs are the first family of receptors responsible for innate immunity, described as transmembrane type 1 proteins attached to the endolysin or plasmatic membrane. The main function of TLR is to recognize the activation mediated by pattern recognition receptors (PRR) of the PAMPs and the DAMPs and induce pyroptosis [34]. TLR in the cellular surface can recognize extracellular pathogens, while TLRs in the lysosomes detect microbial nucleic acids [3,34].

TLRs are encoded in a unique germinal line of PRRs, which participate in the recognition and activation of innate immunity, leading to a cascade if there is inflammation, and release of pro-inflammatory cytokines. These types of receptors are seen in many of the innate immune cells, including macrophages, dendritic cells, T cells, B cells, and natural killer (NK) cells, and in non-immune cells such as tubular epithelial cells, mesangial cells, endothelial cells, and podocytes [3]. The human genome codes about 10 TLRs [1,35]. In DKD, TLR2 and TLR4 have been implicated in the inflammatory aspect of the disease [36].

Among the family of TLRs, TLR4 sends signals via its downstream partner MyD88 to activate NF-kβ, which leads to the production of cytokines and reactive oxygen species (ROS). In hyperglycemia-stimulated podocytes, TLR4 activates NF-kβ and increases the release of cytokines and pro-inflammatory chemokines, contributing to the formation of the inflammasome and progression of the disease [34].

There exists a correlation between the metabolic and inflammatory pathways mediated by TLRs; circulating monocytes have an elevated expression of TLR2 and TLR4, especially in patients with DM type 1 and 2, and this high TLR expression is correlated positively with hemoglobin A1C levels and insulin resistance [36,37]. TLR4 is overexpressed in the tubules and is associated with interstitial infiltration by macrophages in patients with DM type 2 [4]. Its presence is associated with progression of chronic kidney disease (CKD) and correlates negatively with GFR [4].

On the other hand, NLRs are expressed in the cytoplasm and detect intracellular PAMP and DAMP receptors [4]. The NLR family is divided into four subfamilies: NLRA, NLRB, NLRC (including NOD1, NOD2, NLRC3, NLRC4, and NLRC5), and NLRP (including NLRP1-14) [34]. Additionally, the NLRs that make the inflammasome include nucleotide-binding oligomerization domain, leucine-rich repeat, pyrin domain containing 1 (NLRP1), NLRP3, NLRP6, and NLRC4 (C for CARD, caspase activation and recruitment domain), which play a role in the detection of inflammatory disturbances and of the microbiome, which are fundamental in innate inflammatory response [3].

The role of mononuclear renal cells such as macrophages and dendritic cells is of importance, as they contain all the inflammasome and precipitate the activation of the inflammatory cascade. The NLRP3 inflammasome complex is the most well-known and it is implicated in human diseases, such as cancer, hepatic, and renal diseases [38,39]. This complex is made of three components: an adaptor protein ASC (apoptosis-associated speck-like protein containing a caspase-activating recruitment domain) and an effector protein, pro-CASP [3]. Lastly, it is activated by fatty acids, uric acid, uromodulin, extracellular adenosine triphosphate, hyperglycemia, serum amyloid A, and mitochondrial ROS [4].

## 5. The Kallikrein Kinin System (KKS) and DKD

This system and its role in inflammatory processes have been associated with ischemic ictus, wound healing, and cardiovascular and renal diseases [3]. Kininogen is converted into pro-inflammatory peptides known as kinins (bradykinin and kallidin) via tissue or serum kallikrein; the latter are effector pathways [3]. The inhibition of angiotensin-converting enzyme (ACE) can prevent kinin degradation, which would attribute some reno-protective effects to ACE inhibitors. Generally, the KKS system plays a role in inflammation, prevention of apoptosis, and oxidative stress [3].

In the hyperglycemic milieu, there are increased levels of tissue kallikrein, the coagulation enzymes thrombin, trypsin, and factor Xa, and the bradykinin B1 receptor (B1R) [3]. Binding of the bradykinin and kallidin metabolites to B1R induces an NK-Fβ-dependent pro-inflammatory response. Additionally, stimulation of bradykinin B2 receptor (B2R) leads to a pro-inflammatory pathway via activation of MAPK signaling [3]. Tissue kallikrein and other enzymes such as thrombin, trypsin, and factor Xa can stimulate pro-inflammatory and pro-fibrotic pathways by activating the protease-activated receptors (PAR), and given the upregulation of these enzymes in hyperglycemia, it leads to eventual cell proliferation, inflammation, and fibrosis [3].

## 6. Innate Immune Cells

Dendritic cells and macrophages are the effector cells of the innate immune system involved in the inflammatory and the immune response; they are found as infiltrates in kidney tissue of patients with DKD, mediating histological damage and the decrease in GFR [3,4]. The inflammatory response in patients with type 2 DM is seen because of multiple metabolic products derived from prolonged hyperglycemia and hemodynamic changes; among these are ROS, hyperuricemia, and lipid metabolites, which act like DAMP detected by TLRs and NLRs, inducing secondary activation of innate immune cells [40,41].

The overproduction of deleterious metabolic products could affect non-immune renal cells, which would play a role in the release of cytokines and chemokines and in the recruitment of immune cells, as stated in other sections [3]. The proliferation and maintenance of these mononuclear inflammatory cells in the kidney require the presence of colony-stimulating factor-1 (CSF-1), which acts via the c-fms receptor [4]. Therefore, the increased production of renal CSF-1 contributes to the proliferation and activation of key mononuclear renal cells of the innate immune response in patients with DKD [3].

The activation of intercellular adhesion molecule-1 (ICAM-1) and monocyte chemoattractant protein 1 (MCP-1)/C-C motif chemokine ligand 2 (CCL2) is an important pathophysiological event, promoting the migration of immune cells in the diabetic kidney [42,43]. Cells other than macrophages and dendritic cells have been discovered, with markers such as D11b, F4/80, and CD68 described for macrophages, and CD11c, myosin heavy chain II, and CD80/86 for dendritic cells [4]. This activation of intracellular signaling added to the oxidative stress and other factors leads to inflammation and fibrosis, as shown in Figure 1, eventually leading to diabetic kidney disease. 

## 7. Inflammatory Cytokines

Inflammatory cytokines are polypeptide molecules produced by immune cells, endothelial cells, epithelial cells, and fibroblasts in an autocrine, paracrine, and juxtacrine manner [4]. Among these cytokines, IL-1, IL-6, IL-18, (TNF-α), and IL-17 have a strong pro-inflammatory effect in the pathogenesis of DKD [44]. Studies on the role of cytokines in inflammation reveal that IL-1 and its elevation in the urine are associated with epithelial proximal tubular cell and podocyte damage [3].

IL-18 is a member of the IL-1 superfamily, which mediates the stimulation and release of interferon-γ (IFN-γ), second-hand modulating the innate and adaptive immune cells [4]. It has been found to be elevated in the serum and urine of patients with type 2 DM and renal disease, potentially serving as a marker of renal inflammation.

IL-6 works as a co-stimulator and an acute phase reactant, activating T and B cells. It has been found to be elevated in patients with DM type 2 and DKD [45]. The identification of molecules such as IL-6 and IL-10 leads to early diagnosis of DKD, even before the deterioration of eGFR [46].

Tumor necrosis factor-α (TNF-α) is produced by activated macrophages and renal cells in the tubules and the glomeruli [47]. It acts as a pro-inflammatory molecule in DKD; it induces the release of other cytokines and chemokines and promotes apoptosis and cytotoxic effects in general [48]. Its functions include the induction and differentiation of inflammatory cells, renal cytotoxicity, induction of apoptosis, hemodynamic alterations, disruption of glomerular autoregulation, increased endothelial permeability, and production of ROS and oxidative stress [4]. 

Genomic sequencing studies have established an inverse correlation between the expression of (TNF-α) and the drop in eGFR. Several studies in patients with type 2 DM have shown the existing relationship between the renal expression of (TNF-α) and its receptors, associated with a decrease in eGFR and the development of DKD [49,50].

Lastly, the role of IL-17 is fundamental for the development of CD4+ cells that are IL-7+, called 17 helper T cells, which interact with the IL-17 receptor. The dysregulation of IL-17 in autoimmune diseases activates several immunologic pathways that lead to the expression of IL-6, (TNF-α), CCL-2, and CCL-5. Activation of these molecules contributes to the recruitment of monocytes and neutrophils at the site of inflammation [3]. IL-17 levels are particularly elevated in patients with type 1 DM, and they correlate with glomerular filtration. The role of these cytokines correlates with the activation and progression of inflammation, sustaining a close relationship with the remaining complementary pathways and the drop in GFR. Figure 2 shows how the hyperglycemic milieu promotes the release of cytokines in several cells including podocytes, tubular epithelial cells, endothelial and mesangial cells, which eventually lead to inflammation and deposition of extracellular matrix and fibrosis. 

## 8. Chemokines and Their Receptors

Chemokines are small cytokines with the capability of recruiting different cell types via chemotaxis [4]. These proteins are activated in non-immune renal cells as a response to hemodynamic and metabolic changes of diabetic kidneys. This leads to the recruitment migration and adhesion of inflammatory cells [51]. The main chemokines involved in DKD are CCL2 (MCP-1), CCL5 (RANTES), and C-X3-C motif chemokine 1 (CX3CL1, fractalkine), with increasing evidence being published in the literature [4].

CCL-2 is of importance; it is produced by podocytes and renal tubular cells of patients with type 2 DM, which favors the recruitment of mononuclear cells, innate immune cells, and memory T cells on foci of renal inflammation [52]. Elevated urinary CCL-2 levels have been correlated with progression of CKD in patients with and without DKD. Additionally, the degree of inflammatory infiltrate would be correlated with urinary CCL-2 levels, for example, the presence of CD 68+ cells in the interstitium, a marker of inflammatory cells such as macrophages [4].

CCL-5 is involved in the recruitment of monocytes and T cells, playing a central role in the location and margination of leukocytes in sites of inflammation. Elevated CCL-5 levels (RANTES) have been described as a marker of progression towards type 2 DM in patients with obesity and carbohydrate intolerance when compared to healthy subjects [53]. Studies of kidney biopsies in patients with type 2 DM and DKD have been shown to regulate the increase of CCL-2 and CCL-5 levels, which would be related to the progression of renal complications derived from DM. As shown in Figure 2, inflammatory chemokines and cytokines are released in response to hyperglycemia, which eventually leads to inflammation via activation of several signaling pathways, leading to fibrosis. 

## 9. Complement System

Two main complement-mediated mechanisms have been identified in the development of DKD [54]. At first instance, there is the activation of lectin as a response to glycation of proteins that are present in the surface of cells in hyperglycemia [3]. The second mechanism involved is the glycation of complement regulatory proteins as a response to hyperglycemia, with dysfunction of complement regulatory mechanisms. Renal histopathological studies of patients with DKD have demonstrated complement deposits [3].

The complement system is an effector of the innate immune system, with an important role in the pathogenesis of several inflammatory and infectious diseases [4]. There are different components of the complement system in the kidney, especially C3 and C3 convertase in the proximal tubular epithelial cells attached to the membrane, which could activate intrarenal complement in diverse renal diseases [55,56].

Poor metabolic control in hyperglycemia and DKD activates complement mediated by lectin, by the glycation of pattern recognition molecules [57]. These changes lead to the dysregulated activation of the complement system. Glycation of C3 and C4 would not have influence in the progression of the disease. However, the glycation of CD59 leads to the loss of inhibitory function of the membrane attack complex (MAC), leading to endothelial cell damage. CD59 activity is affected by poor glycemic control and is a marker of disease progression [57].

The variation in the concentration of the complement molecules has been associated with an increase in the risk of retinopathy, nephropathy, and diabetic neuropathy, especially high levels of C3 [58]. On the other hand, the concentration of complement proteins such as mannose-binding lectin (MBL) has been associated with an odd’s ratio (OR) of 2.6 for the development of proteinuria and loss of renal function in patients with DM type 2 [59]. Changes in the concentration of MBL could correlate with the progression to end-stage kidney disease and the progression of DKD. Tubular expression of C5a is increased in DKD and is correlated with progression of the disease [60]. The role of innate immunity is fundamental in the role of DKD, with the complement system playing an important role in the initiation and progression. 

## 10. Cells of the Adaptive Immune System

Infiltration of inflammatory cells in the kidneys of diabetic patients is mainly driven by innate immune cells; however, the presence of T cells and B cells is seen in lower proportion [4]. The role of adaptive immune cells has been documented in recent reports; they participate in the pathogenesis of metabolic diseases and are of importance in DKD. In this setting, there is a predominance of the response mediated by Th1 and Th17, with a subsequent reduction in the expression of Treg lymphocytes as an adaptive response to hyperglycemia [61].

In the setting of glomerular disease, the presence of proteinuria in patients with type 2 DM is correlated with an increase in Th1 and Th17 cells [62]. Th1 cells are characterized by the production of cytokines such as IFN-γ, TNF-α, IL-2, and Th17 cells; there is high production of IL-17 [4]. Moon et al. showed an increase in the expression of CD4+, CD8+, and CD20+ in the glomeruli and tubulointerstitium; additionally, CD4+ and CD20+ expression is correlated with the quantity and severity of proteinuria [63]. The role of adaptive immune cells is to be considered in the beginning and during the progression of DKD.

## 11. Aldosterone

The contribution of the inflammation axis known as the aldosterone escape cannot be excluded. In preclinical models, the anti-fibrotic and inflammatory effect of blocking the mineralocorticoid receptor has been documented [64]. This is a current area of active research. Figure 3 shows the role of aldosterone in diabetic kidney disease. 

Aldosterone plays an important role in diabetic kidney disease by increasing the production of extracellular matrix proteins, an important factor in the pathogenesis [65]. Aldosterone induces increased expression of TGF-β1 and the type IV collagen gene under high-glucose conditions in cultured mesangial cells; it also promotes the infiltration of macrophages in the glomerulus and the tubulointerstitium by inducing TGF-β1 expression [65]. Aldosterone also induces other pro-fibrotic molecules that can be possible mediators of diabetic kidney injury, including connective tissue growth factor (CTGF) and plasminogen activator inhibitor-1 (PAI-1), as well as other pro-inflammatory mechanisms via ROS [65]. The presence of aldosterone eventually leads to vascular dysfunction and fibrosis, cardiac hypertrophy and fibrosis, and renal inflammation and fibrosis. Aldosterone also inhibits Treg lymphocytes, which suppress immune response, leading to further dysregulated inflammation and fibrosis. The role of aldosterone in diabetic kidney disease and its association with clinical trials, as well as the other main mechanisms of diabetic kidney disease, are summarized in Table 3.

## 12. Signaling Pathways in DKD

There are several signaling pathways that have been described which are involved in diabetic kidney disease. The TGF-β signaling pathway, a key mediator of fibrosis, exerts its effect via activation of downstream signaling, leading to epithelial-mesenchymal transition (EMT), endothelial mesenchymal transition (endoMT), and myofibroblast activation, prompting a loss of adhesion proteins and connexins in the hyperglycemic milieu, leading to deposition of extracellular matrix molecules [71]. Factors that induce this signaling pathway include elevation of blood glucose and local activation of the renin-angiotensin-aldosterone system (RAAS). The three isoforms, TGF-β1, TGF-β2, and TGF-β3 all have pro-fibrotic effects in renal cells. Downstream of the TGF-β pathway comes the Smad signaling pathway, with Smad3 playing a very important role in renal fibrosis. The activation of the TGF-β/Smad3 pathway is involved in the deposition of collagen [71].

Multiple other pathways downstream of TGF-β have been associated with inflammation and fibrosis in DKD, which are Smad-independent. The mitogen-activated protein kinase (MAPK) pathway, Wnt/β-catenin pathway, extracellular signal-regulated kinase (ERK)1/2, c-Jun N-terminal kinase (JNK), and phosphatidylinositol 3-kinase (PI3K) all crosstalk with TGF-β to promote progression of fibrosis [71].

The MAPK signaling pathway consists of serine/threonine protein kinases, with the main subgroups being P38MAPK, ERK, and JNK. Overexpression of EphA1, a fibrosis modulator, decreased the phosphorylation of ERK1/2 and JNK in the kidney and improved renal fibrosis of mice with diabetic kidneys [71]. Specifically, P38MAPK is involved in the deposition of matrix metalloproteinase 1 (MMP-1), while ERK moderates the production of type 1 collagen in the setting of hyperglycemia. JNK significantly contributes to TGF-β1-induced CTGF mRNA expression, also promoting fibrosis [71].

Hyperglycemia activates the Wnt/β-catenin signaling pathway promoting injury of renal cells with the blockade of this pathway by blocking the LDL receptor, resulting in a reduction of fibrosis [71]. Wnt binds to the frizzled receptor, which interacts with both low-density lipoprotein receptor-related proteins (LRP)5 and LRP6 co-receptors to impede the ubiquitination of destruction complex, therefore leading to inflammation and fibrosis [71]. The EMT process is promoted by the downregulation of E-cadherin expression, with subsequent renal fibrosis [71].

Phosphoinositide 3-kinase (PI3K), an intracellular phosphatidylinositol kinase, and Akt, a protein kinase also known as protein kinase B, is a signaling pathway that may be involved in renal fibrosis. PI3K is initially activated by tyrosine kinase to transform phosphatidylinositol 4,5-biphosphate (PIP2) into phosphatidylinositol 3,4,5-triphosphate (PIP3), therefore leading to the accumulation of Akt, which is then phosphorylated. In hyperglycemia, activation of this pathway may lead to the upregulated expression of TGF-β1, α-SMA, and collagen type 3 [71].

It has been established that the JAK/STAT signaling pathway is associated with renal fibrosis in both humans and mice with diabetic kidneys. A study showed that in hyperglycemic conditions, there is tyrosine phosphorylation of JAK2, STAT1, STAT3, and STAT5, and another study showed that administering a STAT3 inhibitor reduced the pro-fibrotic gene expression of collagen IV, TGF-β1, VEGF, and ACE in tubular epithelial cells in mouse kidneys [71].

The Notch signaling pathway consists of four transmembrane receptors, two JAG-family ligands, and three delta-like ligands [71]. Activation of this pathway was observed to lead to tubulointerstitial fibrosis in mice; additionally, administration of γ-secretase inhibitor, a pharmacological inhibitor of Notch activation, leads to improvement of renal fibrosis in the folic-acid-induced (FA-induced) TIF mouse model, with noticeably decreased levels of collagen, vimentin, and fibronectin [71].

Other proposed signaling pathways include the nitric oxide pathway (NO), which is found in high amounts in diabetic glomeruli and activates the presentation of secreted modular calcium binding protein 1 (SMOC1) such that it stimulates TGF-β and CTGF expression in mesangial cells, therefore inducing fibrosis; other proposed mechanisms are increased levels of plasminogen activator inhibitor-1, which lead to the decreased protease activity of plasmin, resulting in accumulation of ECM proteins [72,73].

## 13. Conclusions

Advances in the knowledge of the pathogenesis of this disease make it clear that it is a systemic inflammatory condition in which the kidneys are highly affected. New pathways involved in the pathophysiology of DKD are proposed regularly, some of which are not clearly described. Inflammation is involved, with genetic and molecular mechanisms implied. Continuous research needs to be performed to continue to understand these factors that affect the progression of the disease.

## Figures and Tables

**Figure 1 ijms-23-08668-f001:**
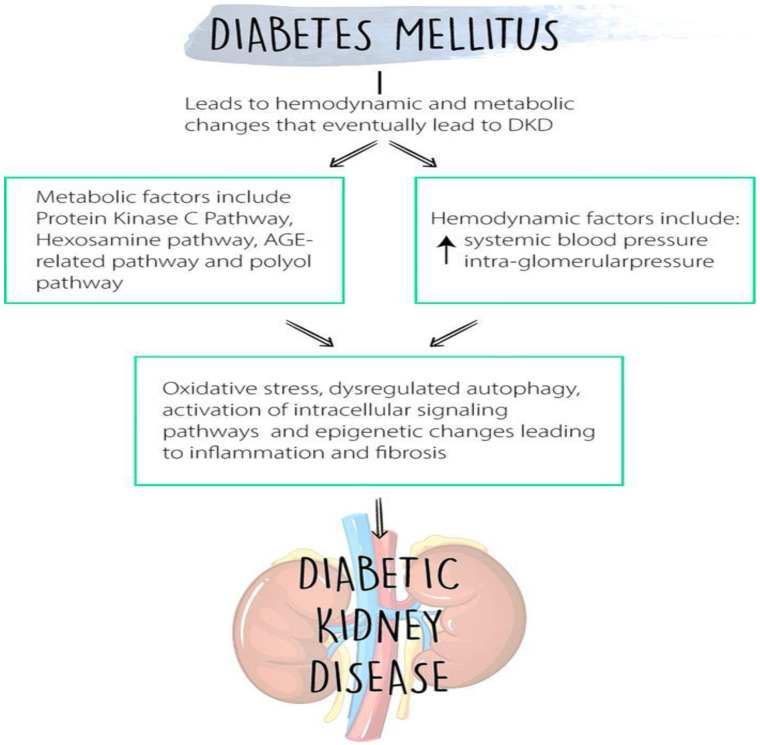
Diabetes mellitus leads to both hemodynamic factors, which include increased blood pressure and intraglomerular pressure, and metabolic factors (including protein kinase C, Hexosamine, polyol and AGE-related pathways leading to ROS and activation of signaling pathways leading to inflammation and DKD.

**Figure 2 ijms-23-08668-f002:**
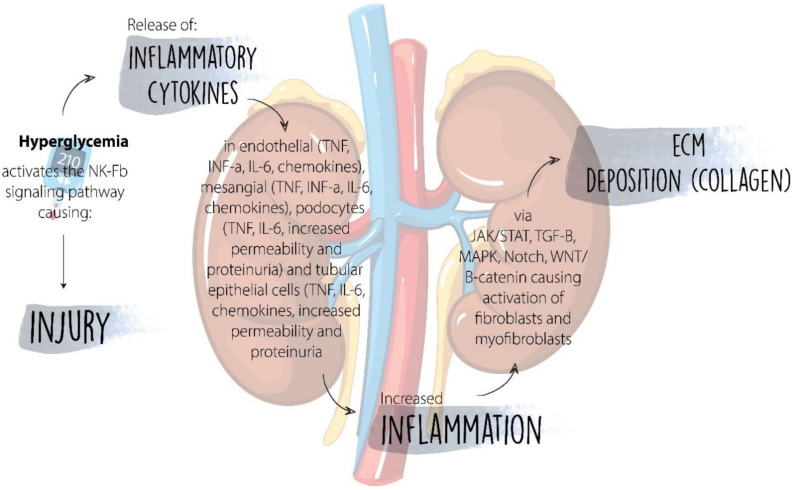
Hyperglycemia leads to the production of pro-inflammatory cytokines and chemokines in several cell types in the kidney including endothelial cells, mesangial cells, podocytes and tubular epithelial cells, causing inflammation via signaling pathways such as the JAK/STAT, TGF- β, Notch among others, leading to fibrosis.

**Figure 3 ijms-23-08668-f003:**
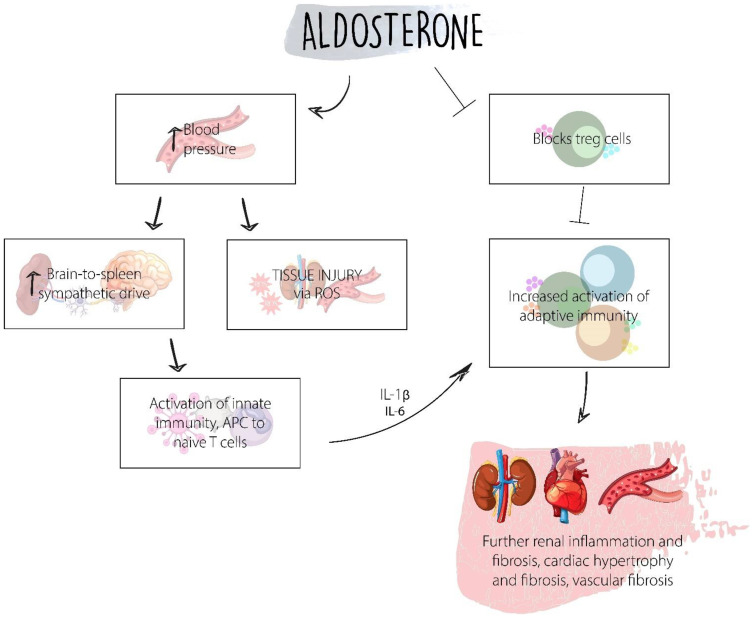
Aldosterone elevated blood pressure leading to tissue injury via ROS, increasing the brain to spleen sympathetic drive, which leads to activation of innate immunity. Cytokines such as IL-6 and IL-1β lead to the activation the adaptive immune system, which leads to further tissue inflammation and fibrosis.

**Table 1 ijms-23-08668-t001:** Animal experiments published to assess biomarkers and their association with kidney diseases or DN [15].

Biomarker (s)	Reference	Population	Study Design	Findings	Potential Clinical Application
Serum SDMA (symmetric dymethylarginine)	Hall et al. (2016) [16]	19 dogs with CKD and 20 control dogs	Retrospective	SDMA detected CKD earlier than creatinine	SDMA as a potential target for evaluation of kidney disease in dogs
Serum SDMA (symmetric dymethylarginine)	Hall et al. (2014) [17]	21 cats with CKD and 21 healthy control cats	Retrospective	SDMA detects CKD earlier than creatinine	SDMA as a potential target for evaluation of kidney disease in cats
Serum SDMA and Cystatin C	Pelander et al. (2019) [18]	30 healthy dogs and 67 dogs with diagnosis or suspicion of CKD	Cross-sectional	Creatinine and SDMA were similar in detecting reduced GFR, whereas cystatin C was inferior	SDMA can be measured together with creatinine for evaluation of kidney function in dogs
β2-microglobulin, calbindin, clusterin, EGF, GST-α, GST-μ, KIM-1, NGAL, osteopontin, TIMP-1, and VEGF	Togashi and Miyamoto et al. (2013) [19]	5 male Zucker diabetic fatty rats (ZDF/CrlCrlj-Leptfa/fa) and 5 male non-diabetic lean rats (ZDF/CrlCrlj-Lept?/+	Cross-sectional	Urinary levels of cystatin C, β2-microglobulin, clusterin, GST-μ, and KIM-1 were increased before the development of histopathological changes consistent with DN	Cystatin C, β2-microglobulin, clusterin, GST-μ, and KIM-1 could be used as markers of DN in mice models of DN
Serum RBP4 (Retinol-binding protein 4)	Van Hoek et al. (2018) [20]	10 cats with CKD, 10 cats with hyperthyroidism, and 10 healthy cats	Cross-sectional	Cats with CKD and hyperthyroidism had higher concentrations of RBP4 than healthy cats	RBP4 can be used as a marker of kidney dysfunction in cats
Plasma NGAL and UNCR	Steinbach et al. (2014) [21]	17 dogs with CKD, 48 dogs with AKI, and 18 control subjects	Cross-sectional	Plasma NGAL concentration and UNCR were significantly higher in dogs with AKI or CKD compared to healthy dogs. In addition, these markers were higher in dogs with AKI compared with dogs with CKD	NGAL is an established marker of AKI, but can also be used to distinguish dogs with CKD from healthy dogs
Urinary KIM-1, NGAL, and vanin-1	Hosohata et al. (2014) [22]	8 male spontaneous type 2 diabetic OLETF rats and 8 male non-diabetic Long Evans, Tokushima Otsuka (LETO) rats	Cross-sectional	Urinary KIM-1 was more sensitive than albumin in detecting DN	KIM-1 can detect early tubular damage in rats with DN

**Table 2 ijms-23-08668-t002:** Clinical trials evaluating biomarkers and their main findings, with potential clinical correlations [23].

Biomarker (s)	Reference	Population	Study Design	Findings	Potential Clinical Application
Plasma endostatin	Carlsson et al. (2016) [24]	607 patients with type 2 DM	Prospective	Edostatin levels are associated with increased risk of GFR decline and mortality	Potential use as a marker of kidney dysfunction in type 2 diabetics
Serum amyloid A	Dieter et al. (2016) [25]	135 patients with type 2 DM	Prospective	Higher levels of serum amyloid A are associated with higher risk of death and ESRD	Amyloid A is a potential target for evaluating diabetic nephropathy in patients with type 2 diabetes
Urinary NGAL and cystatin C	Garg et al. (2015) [26]	91 patients with type 2 DM, 30 patients with prediabetes.	Cross-sectional	NGAL and cystatin C were significantly higher in participants with vs those without microalbuminuria	Early detection of microalbuminuria in patients with type 2 DM
Urinary KIM-1,L-FABP, NAG, and NGAL	Fufaa et al. (2015) [27]	260 patients with type 2 DM	Prospective	NGAL and L-FABP are independently associated with ESRD and mortality	Prediction of ESRD and mortality in patients with type 2 DM
Serum E-selectin, IL-6, PAI-1, sTNFR1, TNFR2	Lopes-Virella et al. (2013) [28]	1237 patients with type 1 DM	Prospective	TNFR1 and TNFR2 and E-selectin are the best predictors of progression to macroalbuminuria	Marker of progression to macroalbuminuria in patients with type 1 DM
Urinary L-FABP	Araki et al. (2013) [29]	618 patients with type 2 DM	Prospective	L-FABP is associated with decline in eGFR	Potential use as a marker of progression of GFR in DN
Plasma TNF-α, TNFR1, and TNFR2, ICAM-1, VCAM-1, PAI-1, IL-6, and CRP	Niewczas et al. (2012) [30]	410 patients with type 2 DM, CKD stages 1-3	Prospective	TNFR1 and TNFR2 were strongly associated with risk of ESRD	Potential use as a marker of progression towards ESRD in patients with CKD stages 1–3 in patients with diabetic nephropathy
Urinary NAGL, NAG, and KIM-1	Fu et al. (2012) [31]	101 T2DM patients, 28 control subjects	Cross-sectional	Every marker showed increased levels in patients with DM; NGAL and NAG were positively correlated with albuminuria; NGAL showed important differences between micro- and macroalbuminuric patients	KIM-1 and NGAL could be potential early markers of DN
β-Trace protein and B2M	Foster et al. (2015) [32]	250 patients with type 2 DM	Prospective	β-Trace protein is associated with ESRD	β-Trace protein is a potential marker for progression towards ESRD in type 2 diabetics
Urinary IL-6, CXCL10/ IP-10, NAG, and KIM-1	Vaidya et al. (2011) [33]	659 patients with type 1 DM, 38 controls	Cross-sectional and prospective	KIM-1 and NAG both individually and collectively were significantly associated with regression of microalbuminuria	Both molecules are potential markers for regression of microalbuminuria in patients with type 1 DM

**Table 3 ijms-23-08668-t003:** Main molecular mechanisms of DKD and their main findings summarized, as well as their association with published clinical trials [47].

Molecular Mechanisms of DKD	Main Findings	Association with Clinical Trials
Genetics and Epigenetics	DNA methylation, histone post-translational modifications, microRNA (miRNA), lcnRNA, and circRNA are exacerbated by hyperglycemia, promoting renal inflammation and fibrosis.	A cohort by van Zuydam et al associated the GABRR1 gene with microalbuminuria [13].
Innate Immunity	Macrophages and dendritic cells are activated by the interaction of TLRs and NLRs by hyperglycemia, leading to inflammation and renal injury. These cells require the presence of CSF-1.	A phase 2 randomized controlled clinical trial showed that baricitinib, a JAK1/2 inhibitor, reduces albuminuria in patients with type 2 DM. The JAK-STAT pathway is important in the initiation and regulation of innate and adaptive immunity [66].
Adaptive Immunity	Seen in lower proportion than innate immune cells; predominantly Th1 and Th17 cells. Increased expression of CD4+, CD8+, and CD20+ cells in DKD.	A phase 2 randomized controlled clinical trial showed that baricitinib, a JAK1/2 inhibitor, reduces albuminuria in patients with type 2 DM. The JAK-STAT pathway is important in the initiation and regulation of innate and adaptive immunity [66].
KKS	In hyperglycemia, the binding of bradykinin and kallidin metabolites to B1R induces an NK-Fβ-dependent pro-inflammatory response. B2R stimulation leads to a pro-inflammatory response via MAPK pathway.	A cross-sectional study by Härma et al measured plasma kallikrein activity in 295 individuals with type 1 DM, showing that lower levels of plasma kallikrein were associated with higher GFRs [67].
Cytokines and Chemokines	Cytokines including IL-1, IL-6, IL-18, (TNF-α), and IL-17 have a strong pro-inflammatory effect in the pathogenesis of DKD; chemokines include CCL2 (MCP-1), CCL5 (RANTES), and C-X3-C motif chemokine 1 (CX3CL1, fractalkine).	Patients with DKD from the CANTATA-SU trial showed that treatment with canagliflozin decreased circulating levels of IL-6, TNF receptor-1, matrix metalloproteinase-7, and fibronectin-1 [68].
Complement System	Activation of lectin and glycation of the complement regulatory protein CD 59 lead to activation of MAC and endothelial cell damage.	
Aldosterone	Induces increased expression of TGF-β1 and type IV collagen in hyperglycemic conditions.	As shown in the FIDELIO-DKD trial and the FIGARO-DKD, finerenone, minelarocorticoid receptor blocker, has positive results in reducing kidney failure/progression and leads to a reduction in cardiovascular morbidity and mortality in patients with DKD [69,70].

## Data Availability

Not applicable.

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
