# Peer review of "Molecular Mechanisms of Diabetic Kidney Disease"

_ijms, 2022, doi:10.3390/ijms23158668_

Round 1
Reviewer 1 Report
In this paper, the authors review molecular mechanisms of DKD. In general, I think the structure of the manuscript is not correct, with several sections containing no information on how the particular ligand/receptor/molecule could be involved in DKD. Some specifics follow:
The section “Genetics and epigenetics of DKD” is far too brief and only has general information. Main genes and pathways identified in GWAS should be cited and discussed. The authors focus their review on inflammatory mechanisms and should therefore explain whether variability in inflammatory genes/routes have been identified as relevant for DKD. The same happens with the Metabolomics and Proteomics section, no specifics whatsoever. In addition, these two sections seem to be out of the general structure of the review, which focus on several proposed mechanisms for DKD
The authors give no information connecting Kallikrein or Aldosterone with DKD, just very vague data on their function. In contrast, other sections such as chemokines, the complement system are well explained.
The review could use a section on the “new pathways” that the authors comment in their conclusions.
Some tables summarizing the main findings for each mechanism would be desirable.
The quality of the figures is poor, please improve.
Author Response
- The genetics and epigenetics section were expanded, with more specific information as suggested, with the mentioning of specific genes in the GWAS that are associated with diabetic kidney disease.
- The section of the association with KKS and DKD was expanded, as suggested by the reviewer.
- The section of aldosterone and its role in DKD was also expanded.
- A section of signaling pathways was added to the manuscript.
- The article was proofread for grammatical errors and corrected.
- A table summarizing the main molecular mechanisms of DKD was added.
- The figures were changed and the quality was improved.
- All changes or additions were highlighted in yellow.
Please attached find manuscript.

Reviewer 2 Report
In this paper Rico-Fontalvo et al. present an interesting review on molecular mechanisms of diabetic kidney disease (DKD), focussing on the inflammatory components of DKD. The work is well organized, with a multitude of references. The figures are well done.
However:
- the section covering epigenetics should be expanded by recent literature (reviewed e.g. in Shao et al. Epigenetics and Inflammation in Diabetic Nephropathy. Front Physiol 12, 649587 (2021).)
- the conclusion of TLR to mainly induce pyroptosis is to narrow (line 91ff). TLR-signaling is important for other forms of regulated cell death to, like apoptosis or necroptosis (Bertheloot et al. Necroptosis, pyroptosis and apoptosis: an intricate game of cell death. Cell Mol Immunol 18, 1106–1121 (2021). https://doi.org/10.1038/s41423-020-00630-3 , Tang et al. The molecular machinery of regulated cell death. Cell Res 29, 347–364 (2019). https://doi.org/10.1038/s41422-019-0164-5).
- the manuscript should be checked for language/spelling by a native speaker. To name only a few:
line 146: "y" => likely "by"
line 147: "and" => likely "hand"
line 151: "colony stimulating -1" => "colony stimulating factor-1"
line 152: "CFS-1" => "CSF-1"
line 166: " polypeptides molecules" => "polypeptide molecules"
Fig. 1 "ang" => "and"
Fig. 2 "an" => "and", "Notck" = "Notch
Author Response
- The language and spelling check that was suggested by the reviewer was done throughout the paper, including the specific ones suggested also highlighted.
- An entire section reviewing specifics of genetics and epigenetics was added to the manuscript, with additional information in several sections of the paper.
- All changes and additions highlighted in yellow.

Round 2
Reviewer 1 Report
The manuscript has been improved
Author Response
In response to the reviewer's comments regarding a minor spell check, a thorough grammar check was done, with several changes made. Attached a copy of the new version.